# A Qualitative Study Exploring the Barriers and Facilitators for Maintaining Oral Health and Using Dental Service in People with Severe Mental Illness: Perspectives from Service Users and Service Providers

**DOI:** 10.3390/ijerph19074344

**Published:** 2022-04-05

**Authors:** Masuma Pervin Mishu, Mehreen Riaz Faisal, Alexandra Macnamara, Wael Sabbah, Emily Peckham, Liz Newbronner, Simon Gilbody, Lina Gega

**Affiliations:** 1Department of Health Sciences, Faculty of Sciences, University of York, Heslington, York YO10 5DD, UK; mehreen.faisal@york.ac.uk (M.R.F.); emily.peckham@york.ac.uk (E.P.); liz.newbronner@york.ac.uk (L.N.); lina.gega@york.ac.uk (L.G.); simon.gilbody@york.ac.uk (S.G.); 2Hull York Medical School, University of York, Heslington, York YO10 5DD, UK; alexandra.macnamara@hyms.ac.uk; 3Faculty of Dentistry, Oral & Craniofacial Sciences, King’s College London, Denmark Hill Campus, Caldecot Road, London SE5 9RW, UK; wael.sabbah@kcl.ac.uk

**Keywords:** mental ill health, oral health, dental health services, health services accessibility, qualitative research

## Abstract

People with severe mental illness suffer from a high burden of oral diseases, which can negatively impact their physical and mental well-being. Despite the high burden, they are less likely to engage in oral health care including accessing dental services. We aimed to identify both the service users’ and service providers’ perspective on the barriers and facilitators for maintaining oral health and dental service use in people with severe mental illness. Qualitative exploration was undertaken using dyadic or one-to-one in-depth interviews with service users in the UK with a diagnosis of schizophrenia, schizoaffective disorder or bipolar disorder. Service providers, including mental health and dental health professionals, and informal carers (people identified as family or friend who are not paid carers) were also interviewed. Thematic analysis of the data revealed three main cross-cutting themes at the personal, inter-personal and systems level: amelioration of the problem, using a tailored approach and provision of comprehensive support. The main barriers identified were impact of mental ill-health, lack of patient involvement and tailored approach, and accessibility and availability of dental services including lack of integration of services. The main facilitators identified were service providers’ effective communication skills and further support through the involvement of carers. The findings suggest that the integration of dental and mental health services to provide tailored support for overall health and well-being, including the oral health of the patient, can better support people with severe mental illness regarding their oral health needs.

## 1. Introduction

Oral health is an important part of general health. Oral diseases affect at least 3.58 billion people worldwide [1]. Oral health affects aspects of social life, including self-esteem, social interaction, job performance and overall quality of life [2]. In addition, oral diseases are associated with other physical health conditions such as diabetes [3] and coronary heart diseases [4]. People with severe mental illness (SMI) have some of the worst health indices and the lowest life expectancy of any section of the UK population [5]. In England the prevalence of people with SMI (patients with schizophrenia, bipolar affective disorder and other psychoses) is 0.95% (all ages) and premature mortality in adults with SMI is 103.6 per 100,000 [6].

The burden of oral disease is particularly high in people with SMI and it remains a largely neglected issue. The evidence shows that oral health among people with SMI is poorer than among the general population. They have a nearly three times higher chance of losing all their teeth (95% Confidence Interval (CI) 1.7–4.6) and higher caries rates (mean difference 5.0, 95% CI 2.5–7.4), compared with people without SMI [7]. Poor oral health has a profound effect on their general health and quality of life [8]. Complications from untreated tooth decay are reported to be a common cause of non-psychiatric hospital admissions among patients with SMI [9,10]. This means that, in addition to the impacts on individual health, oral disease may have implications for health services, including the associated costs of treatment. Oral health interventions have been demonstrated to be cost-effective in relation to children’s oral health [11,12]. It is therefore important to recognise the potential economic benefits of improved oral health amongst those with SMI.

Despite the importance of oral health, maintenance of regular oral hygiene is a challenge for this population. A review by Turner et al. reported that individuals with SMI were significantly less likely to maintain regular tooth brushing in comparison with the general population (OR 0.19, 95% CI 0.08–0.42) [13]. Behavioural risk factors for oral health, such as higher levels of consumption of sugary food and drinks, are also more common amongst those with SMI [14]. Furthermore, despite the high treatment needs in this population, people with SMI are less likely to access dental services and receive routine dental care [13]. This may be due to a lack of motivation, failure to make appointments for routine dental check-ups, inadequate cooperation, poor communication, and social and financial barriers for accessing oral healthcare. In addition, maintaining good oral health can be particularly difficult for this population due to specific challenges, such as side effects of anti-psychotic and anti-depressant medications (e.g., dry mouth) and co-morbidities associated with mental and physical health conditions. Dental anxiety, phobia, symptoms of mental illness and lack of support systems may also contribute to reluctance for dental visiting or maintenance of oral hygiene [15,16].

A phenomenological study exploring oral health experiences and needs among young adults after a first-episode psychosis reported barriers for patients in their general oral care and also related to their experiences with dental professionals with regards to how interventions failed to address their needs [17]. Furthermore, in a narrative review, Slack et al. (2017) investigated individual, organisational and systemic levels of barriers of oral health in people with SMI. The report focuses mainly on downstream individual-level factors. However, the organisational, systemic and policy factors were not covered in depth due to fewer studies focusing on more upstream approaches (organisational and systemic levels) to improving oral health outcomes in this group [15]. To tackle poor oral health, this population needs to be urgently targeted with interventions that are tailored to meet their needs and address the barriers for appropriate oral healthcare at different levels (individual as well as organisational and systemic levels). The recent consensus statement on the 5 years’ action plan to improve oral health in people with SMI also highlighted the importance of understanding the barriers from different stakeholders and suggested more collaborative ‘whole-person care’ approaches [18].

Therefore, it is important to explore the barriers and facilitators of different aspects of oral health care and service use in people with SMI, from the perspectives of both service users and service providers. Thus, in this study we aimed to explore how people with SMI and the related service providers experience and express the barriers and facilitators related to oral health.

## 2. Methods

The present study reports on the qualitative exploration of barriers and facilitators for the maintenance of oral health and dental service use by people with SMI and the views of the service providers. Research ethics approval was obtained from the University of York Health Sciences Research and Governance Committee. All research participants provided written informed consent prior to participating in the study.

### 2.1. Participants and Setting

A convenience sampling technique was employed to recruit service users to the study. People aged over 18 years, living in the UK and having had a self-reported diagnosis of SMI (schizophrenia, schizoaffective disorder or bipolar disorder) were recruited as service users. Health professionals and carers with experience of providing health services to patients with SMI were purposively recruited to allow for a mix of service providers involved in the provision of both dental and mental health care for people with SMI. Participants were recruited through ‘Involvement@York’ which is the patient and public involvement network and resource co-ordinated by the University of York. [19] and social media posts as well as by using current contacts to spread the word about the study. The service providers were also identified from the professional network of the co-investigators. Eligible participants were invited by email to participate in the research and those expressing interest were provided with the information pack and the consent form. An introductory video of the project [20] which was created to provide an overview of the research to aid with recruitment was sent with the invitation email. A mutually convenient time was scheduled for the interviews upon receipt of the signed consent forms.

### 2.2. Data Collection

Based on the preference and convenience of the participants, the in-depth interviews were conducted using either one-to-one or dyadic interview techniques. In line with COVID restrictions in place at the time of the interviews, all interviews were conducted remotely via the video conferencing platform Zoom [21]. The interviews were co-facilitated by MRF and MPM.

A semi-structured topic guide was used to guide the flow of the interviews. For service users, the participants were asked to share their experience of caring for their oral health and use of dental services, any challenges that they faced for doing this and their perception of a best possible service. The service providers were also asked to share their experience of providing care to patients with SMI, their perceptions of challenges that people with SMI face in seeking oral health care and views about the best possible service that could be implemented.

Prior to the interview, a Zoom meeting link was emailed to the participants with guidance on how to join the Zoom meeting. The participants were asked to set aside two hours for the session to allow adequate time for data collection. Participants were given the choice to keep their camera switched off during the interview if they did not feel comfortable speaking with the camera on. The interviewers had their camera turned on through the duration of the interview to allow participants a sense of non-verbal, facial expressions in relation to active and compassionate listening instead of only relying on the interviewers’ tone of voice for these non-verbal cues. All interviews were video recorded by the default recording function in Zoom and participants’ consent was once again sought prior to initiating the recording. As a token of appreciation for their time, all participants were offered a GBP 20 Amazon e-voucher. Upon completion of each interview, the video files were deleted, the audio recordings transcribed verbatim, and transcripts pseudonymised along with the removal of any identifying information. The interviewers wrote down their reflections immediately after the interviews.

### 2.3. Data Analysis

Data analysis was carried out following the thematic analysis procedure described by Braun and Clarke (2006) [22]. In the first step, the two reviewers (MRF and MPM) read the transcripts and discussed them along with their individual reflections. In the second step, coding was performed individually by the two reviewers, which was then discussed to ensure clarity and agreement. Once initial codes were agreed on, they were then collated to form categories and sub-themes. Themes were created by compiling the sub-themes for both service users and the service providers, to identify barriers and facilitators at three levels—personal level, inter-personal level and system level (with thematic maps provided as additional information). NVivo 12 Pro [23] was used for analysing the data.

The data were analysed concurrently to help identify new themes emerging from the data and this also provided an opportunity to add or modify questions to help further explore phenomena during subsequent interviews.

Rigour for the qualitative study process was supported through interviewers’ recording their reflections during or after the interviews and through constant comparison between the accounts of the participants to reduce analysis bias [24].

In addition, in order to obtain validation of our study findings, we conducted 11 one-to-one stakeholder consultations with a diverse range of stakeholders (Head of Research, and Deputy Director of two NHS Foundation Trusts; Professor and Honorary Consultant of Dental Public Health; Director of Research & Clinical Senior Lecturer/Honorary Consultant; Consultant, Health Care Public Health Team, NHS England and NHS Improvement (North East); Associate Research Delivery Manager (NIHR); Peer Consultant and Co-Production advisor, Training Programme Director for Oral Health Improvement and Dental Care Professionals; Physical Health Lead Nurse; Senior Lecturer in Dental Nursing and Dental Hygiene; Member of Oral Health Promotion Team of a NHS Foundation Trusts). We discussed the study and emerging themes on barriers and facilitators to oral health, the data synthesis plan and future recommendations. We took notes on the views and recommendations of the stakeholders during the consultations, which was reflected in the findings and recommendation of the study.

## 3. Results

A total of 17 dyadic and one-to-one interviews were conducted over a period of three months (July–September 2021), with service users and informal carer and service providers. The sessions lasted two hours and allowed time for in-depth exploration of the oral health issues for people with SMI, captured at the service user and the service provider level. The participant demographics are provided in Table 1.

The barriers and facilitators explored were categorised at three levels: (1) the personal level, relating to those barriers and facilitators that the individual service user faced for their oral health care, and the service providers’ perspectives regarding delivery of care; (2) the inter-personal level, indicating those faced at the service user–service provider interface and (3) the system level, for identifying the wider elements and their influence.

The categories compiled were grouped as sub-themes under each of the three levels for both the service users and the service providers (Figure 1 and Figure 2). Analysis of the data revealed cross-cutting themes and for this reason thematic analysis findings for service users and services providers are presented in combination (Table 2).

### 3.1. Theme 1: Ameliorating the Problem

This theme contained sub-themes that linked to the oral-health-related barriers and facilitators that the service users and the providers faced at the personal level.

#### 3.1.1. Impact of Mental Ill-Health

The service users talked about how their mental illness put them at a disadvantage in comparison to the general population. One example of this was a lack of motivation impacting the service users’ ability to maintain good oral hygiene.

*“I mean I can spend days when I can’t actually get out of bed never mind think about cleaning my teeth, you know that’s just not something that’s going to happen.”* (Service user J-06 with diagnosis of bipolar disorder). 

*“I think, when you have a severe mental illness, you can neglect yourself. And a part of that can be you neglect your oral health.”* (Service user H-03 with diagnosis of schizophrenia).

One of the service users also mentioned how her mental health had worsened during the COVID-19 restrictions due to a sense of isolation, which further fed into her apathy for her general well-being and oral health in particular.

*“I mean, at one point I was really good and I was doing everything you know brushing three times a day, using the little inter dental brushes, the mouthwash, I was at the top end of the regime and then my mental health got worse, I think during the second lockdown and that’s when I lost the momentum and I’m struggling to get that momentum back.”* (Service user M-02 with diagnosis of schizophrenia).

Participants also mentioned how their negative life experiences heavily influenced their intention to visit the dentist for regular check-ups or for treatment. These relate to the need to have a sense of trust and rapport with their dental health professional before they could feel comfortable about being under their care.

*“if somebody said that well Hayley (**pseudonym**) can’t look after you today, I drive away for a while, you know, a couple of weeks, if need be, I know she’s moved to another dentist I say where she’s moved to please? because you know, I trust that person, you know I would want to be on her caseload and because it’s an important thing to people who have endured poor mental health and serious mental illness that when they start to trust somebody it becomes a very particular relationship.”* (Service user J-04 with diagnosis of bipolar disorder).

Furthermore, the intrusive nature of dental treatments was also reported as a significant barrier due to associations with past experiences or negative emotions.

*“I think it’s really a common thing like a lot of people have had experiences that you know felt very intrusive and as an invasive and around the mouth, it makes sense to me that, like dentistry is really triggering for that and really replicate some of that feeling of powerlessness feeling of being out of control, it being painful like having to have your mouth open and you’re not in control of that.”* (Service user Sa-07 with diagnosis of bipolar disorder)

In addition, the participants also spoke about the range of emotions that they go through whenever they have to visit a dentist such as shame, fear, anxiety and distress related to dental treatments. It was highlighted that sitting in the waiting room and having to listen to the drilling sounds can provoke anxiety. It was apparent from the interviews that oral health was considered an integral part of general health and well-being and the major barrier that they faced at the service provider’s personal level was lack of understanding about their mental illness and lack of consideration on how to manage the individual patient according to their needs.

*“This level of education is really needed with these groups of individuals around trauma and you know, so that they are psychologically informed and trauma informed. You know who wants to put anybody through any kind of distress, but you know so it’s a group of people that really do need to learn more about their patients.”* (Service user K-05 with diagnosis of schizophrenia and autism).

#### 3.1.2. Having a Positive Attitude

The views of the service users regarding the need for sensitivity and tact while dealing with patients with mental illness were reflected by the service providers as well. They agreed that there was a need to move away from a position of judgment, and to use compassion and empathy when dealing with patients.

*“I think it’s just important not to judge and actually what you think may be normal for a group of patients isn’t and if some of my patients brush maybe once or twice a week, then that’s better than never and that’s actually all I can expect from them. So, I think it’s about being realistic and non-judgmental and starting with basic things…”* (Service provider C-01 working as a community dentist).

The need for the development of effective communication skills in order to be able to effectively communicate with patients who require extra support was also highlighted by the health professionals as an important area that required improvement.

*“So just as much as tooth brushing is a habit, it’s a healthy habit and it needs to be encouraged so again it just comes back to the way in which that conversation happens. It’s not the ‘you need to do it like this’, we need ‘we’re here to educate you and tell you what to do’, it’s more ‘do you understand the benefits of what I am teaching you and can you demonstrate it to me so that I know that you’re able to do it well yourself’ and that’s the approach that I think could go somewhere.”* (Service provider B-10, special care dentist).

#### 3.1.3. Keeping Oral Health on the Agenda

The need for effective communication skills for patient management, highlighted by the health professionals, was further explored to understand how this could be incorporated to address the oral health needs of the service users with SMI. Taking a more holistic approach by considering not just patient’s teeth but the whole person, proportional to their individual needs, was suggested as the way forward.

*“I think that sometimes people may misunderstand that oral health just means mouth and teeth but actually it’s about the whole of the person, including medical but also including and I suppose it’s sort of taking a rounded approach to the person and sort of a holistic approach for that person.”* (Service provider C-01, community dentist).

*“I think education is quite the key and also trying to break down those barriers and say you know we are kind of patient people, we do understand your problems and anxieties and try to find ways of managing that and dealing with that and showing them that it’s not as bad as what they think is.”* (Service provider- H-04, special care dentist).

Training dental professionals in mental health was another area highlighted as needing attention to address the barriers to providing the patients with best possible care.

*“Yes, indeed clinicians don’t tend to raise things if they’re a bit anxious about whether they’re able to deal with what comes up. So, I think there is a need for some mental health training for the dentist. May be even mental health first aid course that can be two days. Not expecting the dentists to train as mental health professionals, that’s a little bit training we have.”* (Service provider- D-07, caring for a person with schizophrenia).

On the other hand, it was also discussed that mental health staff could be trained to look out for their patient’s oral health by flagging up any signs of a problem and referring the patient to receive appropriate care.

*“I think there’s a real awareness now that physical and mental health go hand in hand, and we need to have an angle on both and doesn’t mean you have to be an expert in dentistry in dental hygiene, but just having a general awareness of kind of I don’t know what warning signs or things to look out for. Just making sure, a lot of it might just be making sure people have the regular checks and understanding the importance of that.”* (Service provider S-06, occupational therapist).

### 3.2. Theme 2: Use of a Tailored Approach

Use of a tailored approach was identified as a facilitator at the inter-personal level by both the service users and the service providers.

#### 3.2.1. Need to Be Heard and Understood

The service users felt that they faced discrimination or experienced patronising attitudes because of their impact of mental ill health’, which created barriers for them in accessing dental services.

*“Like the stigma and discrimination around mental health in society generally I think comes into it. People feel anxious that they’re going to be judged and misunderstood and I think that you know, makes it difficult for people, especially like in the acute phase of their illness to sort of make contact with other health providers.”* (Service user Sa-07 with diagnosis of schizophrenia).

The mental health service users expressed their desire to be involved in their treatment planning, to be treated as a whole individual and be given a voice.

*“Mental health, I would say it’s already exploited you know in terms of not giving patients a voice and disabilities can be very life limiting. So, giving people the options and scope around that gives them a strong voice and a recognition that they are involved in their own treatment in healthcare.”* (Service user S-01 with diagnosis of schizophrenia).

#### 3.2.2. Considering the Individual Needs

The service providers, similarly, spoke about how important it is to provide adequate support to people with SMI and the importance of framing the narrative in a way that would involve them more in their own care and decision making.

*“Patient should be provided information about the potential side effects of their medication that they are prescribed, they should have a fully informed choice. So again, that will come under the mental health side of things. I think, historically, some mental health services avoided telling people about all the potential effects because the medications were pretty problematic. Hopefully now that’s pretty much legal and patients in the hereafter provide fully informed consent but I am not sure how thoroughly people still do that”* (Service provider C-05, clinical psychologist).

However, one of the biggest barriers from the health service provider’s point of view was the lack of motivation and lack of compliance on the part of the patient, because even when there was perceived to be adequate support available for a patient, their non-concordance would potentially prevent the service user from benefitting from the dental services.

*“The main thing should just be getting people through the door and to have an examination or to have education about the kind of oral hygiene, that is where there will be the most benefit.”* (Service provider C-05, clinical psychologist).

For this reason, involvement of the carers such as the family or friends was brought up as an important element of managing patients with SMI, not only to motivate them but also to liaise with the health professionals on their behalf.

*“I have had people who don’t want to discuss their trauma or their past and have consented to the person who is supporting them to discuss it and so sometimes having that other person there, it gives them a different way of communicating and if they don’t want to speak about it directly but have allowed their carer or support worker to do on their behalf, that’s also happened sometimes.”* (Service provider C-01, community dentist).

### 3.3. Provision of Comprehensive Support

At the systems level, it was clear from both the service users’ and service providers’ narrative that more comprehensive support is needed to help people with severe mental illness to overcome barriers for their oral health care.

#### 3.3.1. Utilisation of Dental Services

At the service level, there were two main barriers that were identified by the service users: (1) accessibility issues and (2) lack of availability of integrated care.

The service users stated how difficult it was to find an NHS dentist. Even when they were successful in finding one, the dental practice was either too far away, which caused transportation issues, or they would end up being removed from the practice due to missed appointments because of their unstable mental health condition.

*“You know, you have no choice, you know you have to often put your name down, where I live, its centralized system that you can put your name down and then you’ll be allocated a dentist, but it could be somebody on the other side of town to try to keep it local, ‘you know this one’s come up, would you like to register with them?’ and the cost because rather wait another three or four months you are going to say yeah. So then getting across there becomes a problem.”* (Service user J-04, with diagnosis of bipolar disorder).

*“**So the barrier, is the support, if you are unwell how will you be able to get to the appointment? That’s where the barrier is, would there be enough support in order for me to get to the appointment or will I be able to ask questions during the appointment? And if so, will it be with my level of care be affected?* (Service user S-01, with diagnosis of schizophrenia).

The cost of dental treatment was also reported as a significant barrier for seeking dental treatment.

*“Because it’s having access to quality dental care and if it’s costing you 45 quid to go now and a bit of a squirt and clean 45 quid is, you know well that’s Monday, Tuesday, Wednesday, Thursday’s benefits for me well what shall we not pay? Shall we not pay my rent, shall we not pay my council tax; so I am not going see my kids, yeah; no, I am okay with brown teeth and a bit of plaque. You know you’re asking people to make those sort of choices.”* (Service user J-04, with diagnosis of bipolar disorder).

Lack of integration between health services regarding the provision of holistic care and considering the overall health and well-being of the patient was highlighted as the missing piece of the puzzle. The service users reported that this lack of integration meant that oral health was not considered a priority by mental health and other health professionals, without considering the negative impact of poor physical health on their mental health or vice versa.

*“Making every contact count, it does need to be a conversation and part of you know, a multi-disciplinary team approach, social workers, health workers, mental health workers, GPs. You know it’s a bit like the conversation around making sure people get their physical health checks as part of their severe mental illness and medics, I’ve heard them say it before you know ‘we’re not experts in physical health’, but you know what you, you are my consultant psychiatrist, you are my mental health nurse, you are my social worker, you are whoever, you don’t have to be an expert in the field to put in my CPA letter or my discharge letter or the letter to my GP-when was the last time I saw a dentist or when’s the last time I had a physical health check…you know, to advocate for me and that’s what we need, we need people to support us, we need people to advocate for us.”* (Service user K-05, with diagnosis of schizophrenia).

#### 3.3.2. Accessibility and Availability of Services

The dental service providers were aware of the difficulty with finding a dentist but mentioned that due to the way the dental commissioning works and having heavy caseloads, they have to remove a patient from under their care if appointments are frequently missed.

*“Those patients that don’t attend appointments with us, you know they don’t add three hours of our time. So, we are commissioned to deliver those targets, so the practice just you know can’t keep seeing them, you know if they really struggle, unfortunately, to comply with the normal frame of practice in primary care.”* (Service provider E-02, High street dentist).

Getting the dental appointment easily and within a short time was suggested, though such facilities are scarce now.

*“The majority of people who come in, it isn’t that it was their focus or their priority, but if there were any issues there used to be a facility for a very quick referral to a local dentist and the whole system is not there anymore. But it was an NHS dentist and it was possible to bring them in the morning and have an appointment the same day. And that was focused mainly on people who have mental health problems and I think the benefits of that were people got seen straight away, they didn’t have to think about it and the pressing issues, whatever the tooth ache or whatever contentious was fixed straightaway.”* (Service provider- S-08, worked as mental health nurse).

It was also pointed out that with the existing demand for mental healthcare, existing services are at full capacity; therefore, facilitating the patients in other areas of their health might not always be feasible with limited resources.

*“With the caseloads that people carry at the moment you wouldn’t be able to, mental health staff wouldn’t be able to kind of facilitate supporting someone to get those. So even you had that overview and you have that, I don’t know that awareness you still have not got the resources in terms of staffing to be able to support that. And so you, you just continue hitting that barrier, because the people have just got ridiculous caseloads essentially.”* (Service provider M-09, mental health nurse).

However, in line with the service users’ views, the health professionals agreed that there was a lack of integration between services, with every service mostly dealing with one aspect of patients’ health and not working in coordination to improve the overall health and well-being of the patient.

*“So how do we have those conversations about finding a sweet spot for an individual- right balance so that each profession understands the rationale behind what the other one is doing and we’re not always just butting heads, but we’re actually supporting the patient in the middle.”* (Service provider B-10, special care dentist).

## 4. Discussion

The study aimed to explore the barriers and facilitators for the maintenance of oral health and dental service use by people with SMI, informal carers and service providers. Themes were identified and classed at three levels—personal, inter-personal and system levels—to provide an understanding of how the barriers and facilitators can be addressed to provide the best possible care and support through the development and testing of a comprehensive intervention. When discussing with the stakeholders, they agreed with the study findings and recommendations.

A previous qualitative study reported experience of oral health and perceived support by classifying the findings according to five categories: the shame of having poor dental health, history of dental care, experiences of self-care, handling of oral health problems, and experiences of staff support [25]. Similarly, in the current study, we found that the shame of having poor dental health induced feelings of guilt, and a sense of stigma was considered as a major personal-level barrier.

At the interpersonal level, service users stated that lack of sensitivity or a suitable approach by dental service providers while communicating with patients with SMI was a significant barrier for them to access dental services, for example the ‘patronising attitude’ they experienced when accessing services. The service users also highlighted the need for more psychologically and trauma-informed dentists, and more patient involvement in the provision of care. These feelings were also reflected in the accounts of those service users who were very happy with their dental care provider. This was because of the sense of trust and rapport that they shared with them, and the service users expressed fears of losing them due to job changes or re-location. This is in line with the findings reported by Bjørkvik et al. (2021) in their study conducted in Norway to explore perceived barriers for obtaining optimal dental care for patients with SMI. The authors report that patronising attitudes cannot lay the foundation for a respectful relationship between a service user and service provider(s) and that patients should be allowed input in planning their own care provision [26].

In the current study, we also explored the views of different health service providers in relation to barriers and facilitators for oral health maintenance and dental service use by people with SMI. Limitations around access to a dentist including discontinuity of care due to missed appointments and high caseloads were identified as the main system level barriers. These findings are similar to those presented by a study conducted in Australia exploring the views of mental health nurses about dental access by patients with SMI. The authors reported the main barriers related to limited access to dental health care services, the cost of dental treatment and long waiting times [27]. A review also reported that the cost of the care and dental phobia are the most frequently reported barriers to dental care in psychiatric patients. Other barriers included lack of awareness of dental health and mistrust towards dental health providers [16].

The identified facilitators were related to dental professionals’ effective communication skills, the provision of tailored support, the involvement of carers and the need for an integrated care model with interprofessional communication to support the patient’s overall health and well-being and not just one aspect of their health. Similar results were reported by a study conducted in the USA regarding barriers and facilitators for oral health among persons living with mental illness, in which the authors qualitatively explored the views of patients with mental illness, psychiatrists and dentists. The study reported dentists’ chair side manner, community support and interprofessional communication as important professional- and system-level facilitators for supporting the dental needs of patients with SMI [28].

The present study collected data from both service users and service providers. We aimed to recruit service users representative of both male and female genders and covering a range of diagnoses that constitute SMI. Although the number of service users interviewed is relatively small, they are from wide geographical areas, which indicates that the challenges that they face were common in the health system. Within the service providers we included both dental and mental health service providers, as well as an informal carer. The service providers were also from different geographical areas, and the incorporation of the perspective of both these two groups of service providers is a strength of the study. Apart from it being a necessity to conduct interviews remotely at that time, use of videoconferencing to conduct the interviews provided several other advantages. These included the ability to reach participants situated in diverse locations, flexibility in arranging interviews at a convenient time, avoidance of time spent travelling to the venue and associated expenses, and the ability to interview participants in a familiar surrounding whilst preserving the face-to-face aspect of in-person interviews to allow for observation of non-verbal and visual clues [29]. To ensure quality, efforts were put in place to maintain rigour at all times in conducting this qualitative study. We also sought some validation of our findings via the stakeholder consultations. Having that wider stakeholder engagement in this under-researched area was an important addition to the validity of our findings.

There are some limitations of the study. One main limitation was the small sample size. Within the remit of the study, we were able to recruit a limited number of service users (seven service users, nine health professionals and one informal carer). We were able to recruit only one informal carer who showed an interest in taking part within the study recruitment phase. The recruitment of both informal and paid carers could bring their perspective more comprehensively. There was also a lack of recruitment from diverse ethnic backgrounds. This would have increased the possibility of inclusion of cultural and ethnic diversity among participants and of different viewpoints, thereby further enriching the data. We did not collect data about participants’ socio-economic conditions. Allowing for more recruitment time and contacting related gatekeepers and collecting socio-demographic information of the participants should be considered for a similar future study. Nonetheless, this study is an important addition to the knowledge base, as this study highlighted an important and under-researched topic. Within the constraints of time and resources, our good quality but modestly sized study brings to light some very important issues which need to be highlighted. Further research is needed with a larger and diverse sample to explore the views on specific aspects of interventions to improve oral health in this population, which is important for developing or adapting effective interventions/programmes for improving the oral health and daily self-care of people living with SMI.

## 5. Conclusions

In this qualitative study the main barriers identified were the impact of mental ill-health, the lack of patient involvement and a tailored approach, and issues aroundaccessibility and availability of dental services, including the lack of integration of services. The main facilitators identified were service providers’ effective communication skills and further support through the involvement of carers. Our findings suggest the need for a comprehensive approach to better support people with SMI in their oral health care needs. Integration of services and provision of tailored support with a focus on the overall health and general well-being of the patient has been highlighted as the most important next step by both the service users and the service providers.

## Figures and Tables

**Figure 1 ijerph-19-04344-f001:**
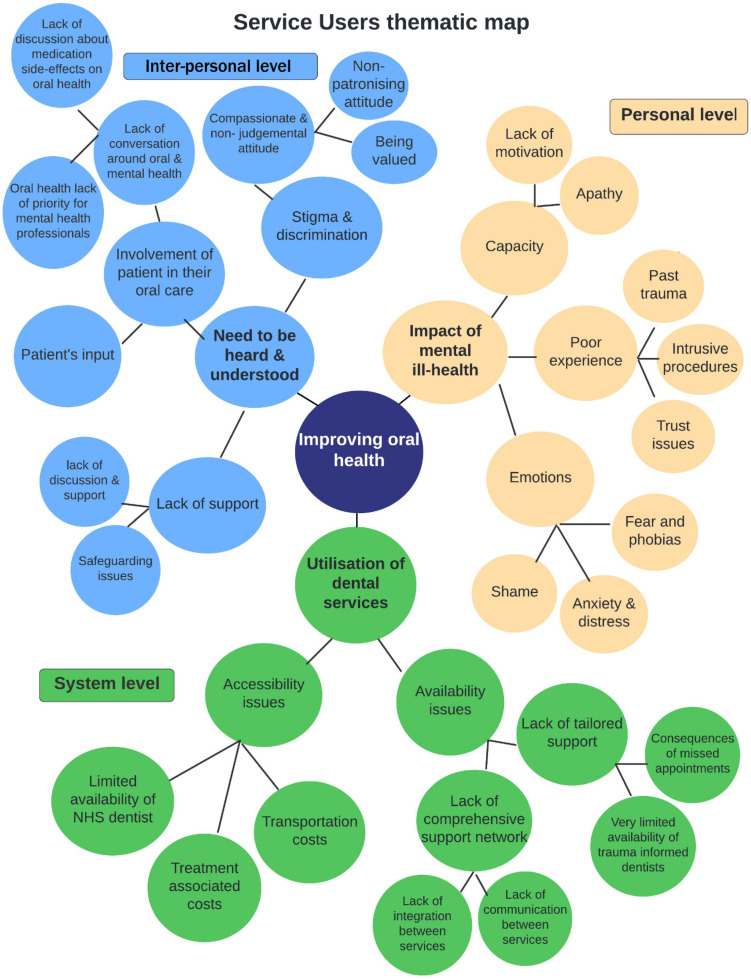
Thematic map of service users’ perspectives on barriers to improving oral health in people with severe mental illness at personal, inter-personal and system levels.

**Figure 2 ijerph-19-04344-f002:**
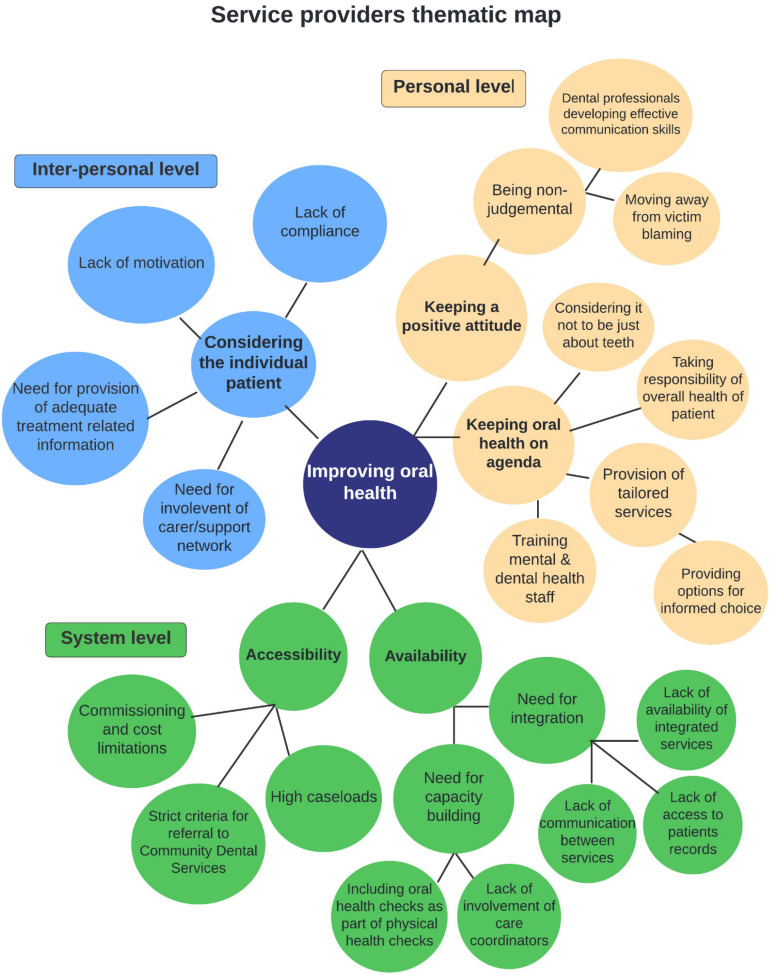
Thematic map of service providers’ perspectives on barriers and recommendations for improving oral health in people with severe mental illness at personal, inter-personal and system levels. CDS: Community Dental Services.

**Table 1 ijerph-19-04344-t001:** Demographics of the study participants.

ID	Participant Group	Age (Years)	Gender	Diagnosis/Profession
1	Service user	31–40	M	Schizophrenia
2	Service user	>60	F	Schizophrenia
3	Service user	31–40	F	Schizophrenia
4	Service user	41–50	M	Schizophrenia
5	Service user	41–50	M	Bipolar disorder
6	Service user	>60	F	Bipolar disorder
7	Service user	41–50	F	Bipolar disorder
8	Health professional	31–40	F	Community service dentist
9	Health professional	31–40	F	High street dentist
10	Health professional	31–40	F	Dental hygienist
11	Health professional	31–40	M	Special care dentist
12	Informal Carer	51–60	F	Caring for person with schizophrenia
13	Health professional	31–40	M	Occupational therapist
14	Health professional	31–40	M	Clinical psychologist
15	Health professional	31–40	F	Mental health nurse
16	Health professional	41–50	F	Mental health nurse
17	Health professional	31–40	M	Special care dentist

**Table 2 ijerph-19-04344-t002:** Themes with associated levels, sub-themes and categories.

**Level- Theme**	**Perspectives of Service User/Service Provider**	**Sub-Themes**	**Categories**	**Description**	**Reference to Theme (n)**
Personal level- Amelioration of the problem	Service user	Impact of mental ill-health	Capability Emotions Negative experience	Problems associated with mental illness	15
	Service provider	Having a positive attitude	Being non-judgemental	Incorporation of effective communication skills	3
	Service provider	Keeping oral health on the agenda	Taking responsibility of overall health Provision of tailored services Considering it to be not just about teeth Training mental and dental health staff	Taking a holistic approach	15
Inter-personal level-Use of a tailored approach	Service user	Need to be heard and understood	Stigma and discrimination Lack of patient’s involvement Lack of support	Patient involvement in care provision	27
	Service provider	Considering individual needs	Lack of compliance Lack of motivation Need for provision of adequate treatment-related information Need for involvement of carer/support network	Provision of care proportional to needs	4
Systems level-Provision of comprehensive support	Service user	Utilisation of dental services	Accessibility issues Availability issues	Factors affecting service utilisation	52
	Service provider	Accessibility and availability of services	Commissioning and cost limitations Community Dental Services referral criteria High caseloads Need for integration Need for capacity building	Consideration for integration of services	46

## Data Availability

No new quantitative data were created or analysed in this study. Data sharing is not applicable to this article. The related transcripts from where the quotes were used in this article are available upon sensible request from the corresponding author.

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
