# Peer review of "A Qualitative Study Exploring the Barriers and Facilitators for Maintaining Oral Health and Using Dental Service in People with Severe Mental Illness: Perspectives from Service Users and Service Providers"

_ijerph, 2022, doi:10.3390/ijerph19074344_

Round 1

Reviewer 1 Report

The topic of the manuscript is the service users’ and service providers’ perspectives on the barriers and facilitators for maintaining oral health and dental service use in people with severe mental illness.

The title and the abstract of the article are informative. The Introduction presents the issue of individual, organisational and systemic levels barriers of oral health in people with severe mental illness. The section "Material and Methods" precisely explains the chosen study design. However, the section "Results" should be improved. The Discussion is interestingly written, including the short paragraph about the study limitations. The Conclusions seems to be the "take-home" messages.

Some following points must be clarified/corrected for the further processing of this article.

Merits-related comments:

  1. In the Abstract almost no reference to the Results section – this needs to be supplemented.
  2. Please complete keywords with the proper MeSH terms, necessary for indexing in the databases.
  3. This tiny sample size is completely incomprehensible for studies of patients with relatively frequent mental disorders carried out remotely via Zoom – only 7 service users. So the number of observations should be increased. Even if the authors try to call their study as a pilot/preliminary study. In contrast, the group of service providers is very heterogeneous and includes a variety of medical professionals. This element of the study design should be better organised.
  4. The Results section contains only subjective excerpts from the respondents' statements. Certainly, more is required for scientific publications, e. g. a quantitative analysis of determining factors, not just a description of the statements of some subjects. Therefore, the Results part requires considerable editorial and methodological revisions. It is proposed to be enriched with summary tables and easy-to-read new figures.
  5. It is suggested to add more recent articles from 2019-2021 to the references in the Introduction and the Discussion.
  6. At the end of the Discussion, the potential limitations of the study should be explained more clearly, how they are justified and how they can be addressed in other possible studies. There are certainly more limitations than those mentioned so far.
  7. Conclusions may also require rewording after modification of the Results part.

Technical comments:

  1. The size of fonts in Figures 1 and 2 should be larger. In its current form, the text is unreadable.
  2. In the text, reference numbers should be placed in square brackets [ ], and placed before the punctuation; for example [1], [1–3] or [1,3].
  3. The citation list must be corrected. References should be described as follows:
    1. Author 1, A.B.; Author 2, C.D. Title of the article. Abbreviated Journal Name YearVolume, page range.
  4. In Author Contributions, the following statements should be used "Conceptualization, X.X. and Y.Y.; Methodology, X.X.; Software, X.X.; Validation, X.X., Y.Y. and Z.Z.; Formal Analysis, X.X.; Investigation, X.X.; Resources, X.X.; Data Curation, X.X.; Writing – Original Draft Preparation, X.X.; Writing – Review & Editing, X.X.; Visualization, X.X.; Supervision, X.X.; Project Administration, X.X.; Funding Acquisition, Y.Y.".

Author Response

Thank you very much for your valuable suggestion and comments. The response and the changes made in response to the comments are as follows.

  1. 1. In the Abstract almost no reference to the Results section – this needs to be supplemented.

Response: Following brief details about the results now have been added (line 24-27):

The main barriers identified were impact of mental ill-health, lack of patient involvement and tailored approach, and accessibility & availability of dental services including lack of integration of services. The facilitators identified were service provider’s effective communication skills and further support through involvement of carers.

  1. 2. Please complete keywords with the proper MeSH terms, necessary for indexing in the databases.

Response: Keywords as MeSH terms as listed below have now been added (line 31-32)

Mental ill-health, oral health, dental health services, health services accessibility, qualitative research.

  1. 3. This tiny sample size is completely incomprehensible for studies of patients with relatively frequent mental disorders carried out remotely via Zoom – only 7 service users. So the number of observations should be increased. Even if the authors try to call their study as a pilot/preliminary study. In contrast, the group of service providers is very heterogeneous and includes a variety of medical professionals. This element of the study design should be better organised.

Response: Thank you for your comments. The sample size is what feasible within the remit of this project and in the given time frame. The idea of qualitative research is not to generalise statistically significant findings but to give a voice to people so they can share their experience. Furthermore, there is now a shift towards moving away from the unattainable concept of ‘saturation’, hence, the study did not strive for saturation but considered reporting experiences of people who have had a lived experience of mental illness and who were willing to share their experiences. Within the remit of the study, we recruited service users representative of both male and female genders and covering a range of diagnoses that constitute SMI.  

We have now added the following paragraph on stakeholder consultation that we conducted after finishing the interviews, which we thought would strengthen the study.

In Methods we added:

‘In addition, we conducted 11 one to one stakeholder consultations with a diverse range of stakeholders (Head of Research, and Deputy Director of two NHS Foundation Trusts; Professor and Honorary Consultant of Dental Public Health; Director of Research & Consultant, Health Care Public Health Team, NHS England and NHS Improvement (North East); Associate Research Delivery Manager (NIHR); Peer Consultant & Co-Production advisor, Training Programme Director for Oral Health Improvement & Dental Care Professionals; Physical Health Lead Nurse; Senior Lecturer in Dental Nursing & Dental Hygiene; Member of Oral Health Promotion Team of a NHS Foundation Trusts). We discussed the study and emerging themes on barriers and facilitators to oral health, data synthesis plan and future recommendations.’

In the Discussion we added the sentence: ‘The study findings were agreed with the stakeholders.’

As we wanted to ensure the representatives of both dental, mental and informal carers, therefore included this heterogeneous group of service providers. We made this point clear in the Section 2.1 Participants and setting-

‘ Health professionals and carers with experience of providing health services to patients with SMI were purposively recruited to allow a mix of service providers involved in provision of both dental and mental health care for people with SMI’

We consider this as a strength of our study. In the Discussion we mentioned-

‘Within the service providers we included both dental and mental health service providers, as well as an informal carer.’

  1. 4. The Results section contains only subjective excerpts from the respondents' statements. Certainly, more is required for scientific publications, e. g. a quantitative analysis of determining factors, not just a description of the statements of some subjects. Therefore, the Results part requires considerable editorial and methodological revisions. It is proposed to be enriched with summary tables and easy-to-read new figures.

Response: Thank you for your comments. In the Methods we mentioned that it is a qualitative study: ‘The present study reports on the qualitative exploration of barriers and facilitators for maintenance of oral health and dental service use by people with SMI and views of the service providers.’

It is common practice in reporting of qualitative research to just focus on participant’s voices presented as verbatim quotes. As our study employed thematic analysis, we did not think it necessary to provide quantitative analysis in addition to the qualitative findings. Quantification of results such as number of respondents referring to a theme is more commonly carried out when conducting content analysis, but in light of the reviewers comments we have provided a summary table (Table 2).

  1. 5. It is suggested to add more recent articles from 2019-2021 to the references in the Introduction and the Discussion.

Response: More of the recent articles (referenced below) now added to the introduction.

Kuipers, S.; Castelein, S.; Malda, A.; Kronenberg, L.; Boonstra, N. Oral health experiences and needs among young adults after a first-episode psychosis : a phenomenological study. Journal of Psychiatric and Mental Health Nursing 2018, 25, 475-485, doi:https://doi.org/10.1111/jpm.12490.

The Right to Smile; an Oral Health Consensus Statement for People experiencing Severe Mental Ill Health, Closing the Gap Network, 2022.

  1. 6. At the end of the Discussion, the potential limitations of the study should be explained more clearly, how they are justified and how they can be addressed in other possible studies. There are certainly more limitations than those mentioned so far.

Response: More details now added as part of limitations in the discussion (line 472-485) and as part of research recommendations.

‘There are some limitations of the study. One main limitation was small sample size. Within the remit of the study, we were able to recruit a limited number of services users (seven service users, nine health professionals and one informal carer). We were able to recruit only one informal carer who showed interest to take part within the study recruitment phase. Recruitment of both informal and paid carer could bring their perspective more comprehensively. There was also lack of recruitment from diverse ethnic background. It would have increased the possibility of inclusion of cultural and ethnic diversity among participants and including different viewpoints, thus further enriching the data. We did not collected data about participants’ socio-economic conditions. Allowing more recruitment time and contacting related gate keepers and collecting socio-demographic information of the participants should be considered for similar future study. Further research is needed with a larger and diverse sample to explore the views on specific aspects of interventions to improve oral health in this population, which is important to develop or adapt effective interventions/programmes for improving oral health and daily self-care for people living with SMI’

  1. 7. Conclusions may also require rewording after modification of the Results part.

Response: It is not possible to amend the results at this stage, hence we added detailed limitation of small sample in the discussion section. However, we have reworded the conclusion to make the main study findings clear-

‘In this qualitative study the main barriers identified were impact of mental ill-health, lack of patient involvement and tailored approach, and accessibility & availability of dental services including lack of integration of services. The main facilitators identified were service providers’ effective communication skills and further support through involvement of carers. Our findings suggest the need for a comprehensive approach to better support people with SMI in their oral health care needs. Integration of services and provision of tailored support with a focus on the overall health and general wellbeing of the patient has been highlighted as the most important next step by both the service users and the service providers.’

Technical comments:

  1. 1. The size of fonts in Figures 1 and 2 should be larger. In its current form, the text is unreadable.

Response: To increase the fonts of the figures, we are requesting the editor of the journal to put this pictures in a single page in a Landscape orientation and under the text of a page.

  1. 2. In the text, reference numbers should be placed in square brackets [ ], and placed before the punctuation; for example [1], [1–3] or [1,3].

Response: The in-text citation style has now been amended.

  1. 3. The citation list must be corrected. References should be described as follows:
  2. Author 1, A.B.; Author 2, C.D. Title of the article. Abbreviated Journal Name Year, Volume, page range.

Response: The reference list has now been amended.

  1. In Author Contributions, the following statements should be used "Conceptualization, X.X. and Y.Y.; Methodology, X.X.; Software, X.X.; Validation, X.X., Y.Y. and Z.Z.; Formal Analysis, X.X.; Investigation, X.X.; Resources, X.X.; Data Curation, X.X.; Writing – Original Draft Preparation, X.X.; Writing – Review & Editing, X.X.; Visualization, X.X.; Supervision, X.X.; Project Administration, X.X.; Funding Acquisition, Y.Y.".

Response: The Author Contributions has now been amended as suggested.

Reviewer 2 Report

Well don on a well put together study and paper

It would be of value if you can comment on why there was only 1 carer in the sample

Author Response

Thank you for your encouraging comments. The reason of recruiting only one carer has now been clearly mentioned as part of the limitations (line 474-477)

‘There are some limitations of the study. One main limitation was small sample size. Within the remit of the study, we were able to recruit a limited number of services users (seven service users, nine health professionals and one informal carer). We were able to recruit only one informal carer who showed interest to take part within the study recruitment phase. Recruitment of both informal and paid carer could bring their perspective more comprehensively. There was also lack of recruitment from diverse ethnic background. It would have increased the possibility of inclusion of cultural and ethnic diversity among participants and including different viewpoints, thus further enriching the data. We did not collected data about participants’ socio-economic conditions. Allowing more recruitment time and contacting related gatekeepers and collecting socio-demographic information of the participants should be considered for similar future study. Further research is needed with a larger and diverse sample to explore the views on specific aspects of interventions to improve oral health in this population, which is important to develop or adapt effective interventions/programmes for improving oral health and daily self-care for people living with SMI’

Reviewer 3 Report

A novel research about a very difficult subject. The manuscript is very well presented.

Author Response

Thank you for reviewing the paper and for your encouraging comments. 

Reviewer 4 Report

This manuscript addresses the facilities and barriers to dental care faced by both health care providers and patients with severe mental illness. The authors use qualitative research as a strategy to achieve the objectives. 
The article is coherent, well written, and provides valuable data of a possible immediate clinical application. 
One weakness is the small number of participants, which the authors note in the Discussion section. 
The manuscript does not require any corrections, and I have no suggestions; in my opinion, it is an excellent qualitative study.    

Author Response

Thank you for reviewing the paper and for your encouraging comments. We hope the study will be able to illustrate the barriers and facilitators for maintaining oral health and dental service use by the service users and service providers to a wide range of audiences.

Round 2

Reviewer 1 Report

The Authors have made changes by referring to most of the comments. I understand that this is a qualitative study, but so it is relatively subjective. In particular, given the small sample size and the diversity of respondents involved, it is difficult to make the conclusions too clear.

The Authors described the indicated limitations of the study, however, they do not fully convince me of the validity of such a small sample size. Suppose the only factor was the three-month period of the study. Would it not be reasonable to extend this time period and conduct a higher-quality study before publishing the results in a foreign journal? Or maybe the study design should be marked as preliminary? This tiny study group is a severe source of potential research biases.

Besides, I have no new comments. The manuscript still requires editorial revisions (regarding the style of citations in the text and their list, figure fonts, etc.).

Author Response

We would like to thank the peer reviewer for the comments and concern related to the sample size. Whilst this is a small study, which was constrained by time and resources, it has been conducted in a rigorous manner. We have two experienced qualitative researchers (LN and MF) in the team who worked closely to design the methodology of the study that was carefully reviewed by the other senior members of the team (as in co-author list)

In response to the comment which remains focussed on the sample size and the concerns regarding validity of findings and potential bias, we like to state that the results have been discussed in relation to other studies in the Discussion section that found to be consistent with other studies with some detailed addition. We have sought some validation via the stakeholder consultations that has now been added in the manuscript. Having that wider stakeholder engagement on this under-researched topic is considered as an important addition to the validity of findings. We made that point clear in the manuscript (please see lines in Methods:156, 165-166 and in the Discussion: 477-479)

We would also like to highlight that although the number of service users interviewed is relatively small they are from a wide geographical areas which indicates the challenges that they face were common in the health system. Furthermore, the concept of external validity in quantitative research relates to transferability, so having participants from various locations in the country suggests that the findings are transferable to other similar settings. Moreover, the service providers who are engaged with both dental and mental health care were also from different geographical areas and incorporation of the perspective of both these two groups (dental and mental) of service providers is a strength of the study. We made that point clear in the manuscript (please see lines in the Discussion: 464-479)

In relation to the comment on ‘conducting a higher-quality study before publishing the results’, we want to reiterate that in qualitative research bigger sample does not equate to higher quality. The quality of study in qualitative research refers to the rigour with which the study has been performed. Efforts were put in place to maintain rigour at all times in conducting this study. Moreover, what is most important is that this study brings to light some very important issues which need to be highlighted. As this study is on an important and under-researched topic so our good quality but modestly sized study is an important addition to the knowledge base. We made that point clear in the manuscript (please see lines in the Discussion: 492-495)

The study was funded through a small pump priming grant and for a one-year time, which is nearly close to an end. It would be beyond the capacity of the team to extend the project time and recruit more participants to conduct more interviews.

Regarding the Font size of the Figures, we have requested the editor to put the Figures in a single page so that it will be bigger. We have edited the style of citations in the text (for reference no 22).